# Phenotypic Investigation of Virulence Factors, Susceptibility to Ceragenins, and the Impact of Biofilm Formation on Drug Efficacy in *Candida auris* Isolates from Türkiye

**DOI:** 10.3390/jof9101026

**Published:** 2023-10-19

**Authors:** Ozlem Oyardi, Elif Sena Demir, Busra Alkan, Selda Komec, Gonca Erkose Genc, Gokhan Aygun, Leyla Teke, Deniz Turan, Zayre Erturan, Paul B. Savage, Cagla Bozkurt Guzel

**Affiliations:** 1Department of Pharmaceutical Microbiology, Faculty of Pharmacy, Gazi University, Ankara 06330, Türkiye; 2Department of Pharmaceutical Microbiology, Faculty of Pharmacy, Istanbul University, Istanbul 34116, Türkiye; elifsenademir@gmail.com (E.S.D.); busraalkan17@gmail.com (B.A.); cagla.bozkurt@istanbul.edu.tr (C.B.G.); 3Laboratory of Medical Microbiology, Basaksehir Cam and Sakura City Hospital, Istanbul 34480, Türkiye; selda.komec@saglik.gov.tr; 4Department of Medical Microbiology, Istanbul Faculty of Medicine, Istanbul University, Istanbul 34093, Türkiye; gonca.genc@istanbul.edu.tr (G.E.G.); zerturan@istanbul.edu.tr (Z.E.); 5Department of Medical Microbiology, Cerrahpasa School of Medicine, Istanbul University-Cerrahpasa, Istanbul 34098, Türkiye; gokhan.aygun@iuc.edu.tr; 6Clinic of Microbiology, Gaziosmanpasa Training and Research Hospital, University of Health Sciences, Istanbul 34255, Türkiye; leyla_teke@hotmail.com; 7Medical Microbiology Laboratory, Haydarpasa Numune Training and Research Hospital, University of Health Sciences, Istanbul 34668, Türkiye; dennizturan@hotmail.com; 8Department of Chemistry and Biochemistry, Brigham Young University, Provo, UT 84602, USA; pbsavage@chem.byu.edu

**Keywords:** *Candida auris*, ceragenins, virulence

## Abstract

*Candida auris* has emerged as a significant fungal threat due to its rapid worldwide spread since its first appearance, along with its potential for antimicrobial resistance and virulence properties. This study was designed to examine virulence characteristics, the efficacy of ceragenins, and biofilm-derived drug resistance in seven *C. auris* strains isolated from Turkish intensive care patients. It was observed that none of the tested strains exhibited proteinase or hemolysis activity; however, they demonstrated weak phospholipase and esterase activity. In addition, all strains were identified as having moderate to strong biofilm formation characteristics. Upon determining the minimum inhibitory concentrations (MIC) of ceragenins, it was discovered that CSA-138 exhibited the highest effectiveness with a MIC range of 1–0.5 µg/mL, followed by CSA-131 with a MIC of 1 µg/mL. Also, antimicrobial agents destroyed mature biofilms at high concentrations (40–1280 µg/mL). The investigation revealed that the strains isolated from Türkiye displayed weak exoenzyme activities. Notably, the ceragenins exhibited effectiveness against these strains, suggesting their potential as a viable treatment option.

## 1. Introduction

There has been a rise in reports detailing invasive infections caused by multidrug-resistant *Candida* spp. worldwide. One particular species, *Candida auris*, has gained significant attention due to its association with widespread outbreaks of severe infections in healthcare facilities [1]. Prior to 2009, *C. auris* was previously regarded as an infrequently encountered microorganism. After *C. auris* was first isolated from a patient’s ear canal infection in Japan in 2009 [2], it has been reported at an increasing rate from various sites worldwide [3]. According to whole sequencing studies, *C. auris* isolates have been classified into four major clades: South Asia strains (clade I), East Asia strains (clade II), South Africa strains (clade III), and South America strains (clade IV). A 2018 study conducted in Iran found that the *C. auris* isolate from a girl who had not previously traveled abroad potentially belonged to a fifth clade [4]. *C. auris* has caused outbreaks in numerous nations, with notable incidence rates documented in European countries, including Spain, the United Kingdom, and Italy. According to the report by The European Centre for Disease Prevention and Control (ECDC), *C. auris* case numbers between 2020 and 2021 were double those of previous years [5]. Similarly, since it was first announced in the United States in 2016, the incidence of *C. auris* appears to have accelerated significantly in recent years. Additionally, the same report highlights that the cases of echinocandin resistance have increased significantly over the same period [6]. While Türkiye had not reported any cases of *C. auris* until 2020, the first case of this infection was documented in 2020 [7].

*C. auris* causes prolonged hospitalization of immunocompromised patients for various reasons, in addition to high mortality rates, imposing significant burdens on the healthcare system [8,9]. Hu et al. reported that men and elderly patients were more prone to *C. auris* infections, and the mortality rate was 47.5%. Moreover, the main mortality reasons were determined as sepsis, septic shock, and organ failure, especially kidney disease [3]. The ability of *C. auris* to colonize living and non-living surfaces for a long time causes epidemics in hospitals, and studies have shown the difficulty of eradicating this pathogen from the hospital environment [3]. The correct management of *C. auris* infections primarily relies on accurately detecting the causative agent. *C. auris* is difficult to identify by standard laboratory methods and may lead to misdiagnosis. For this reason, it should be defined at the species level in laboratories equipped with the required capacities. In addition, determining the antifungal resistance profile and choosing the right antifungal agent are essential for proper treatment. For pan-resistant strains, combination therapy or experimental drugs may be required. Also, one of the other critical steps in correctly managing these infections is to prevent the pathogen’s spread in hospitals by choosing a suitable disinfectant [10].

One of the most important reasons for the high prevalence and mortality of *C. auris* infections is widespread antifungal resistance and virulence factors such as biofilm formation and exoenzyme production. Studies have shown that *C. auris* has the ability to form biofilms on biotic surfaces such as skin and non-biotic surfaces such as medical devices. Biofilm formation causes both increased resistance of *C. auris* against antifungals and the body’s defense systems and makes it more difficult to eliminate this pathogen from hospitals [11]. In addition, the presence of exoenzymes such as phospholipase, esterase, proteinase, and hemolysin may contribute to the pathogenesis of *C. auris* by facilitating its invasion and penetration into host tissues. Therefore, to better understand the pathogenesis of *C. auris*, it is essential to determine the virulence factors of the strains. It has become a priority to control and treat due to its potential to cause outbreaks in healthcare settings and its antibiotic resistance. Therefore, researchers are concentrating on finding new antimicrobials to combat this fungus [12].

The ceragenins (cationic steroid antibiotics-CSA) tested in this study as new antimicrobials are bile acid-based compounds designed to mimic the activity of endogenous antimicrobial peptides (AMPs). The nature of ceragenins provides a means to avoid problems that arise in the large-scale production and stability of AMPs. Ceragenins (e.g., CSA-13, CSA-131, and CSA-192) are highly active against Gram-negative and Gram-positive bacteria, including against organisms displaying antimicrobial resistance [13,14,15]. Also, ceragenins have been reported to be effective against biofilm forms of bacteria and fungi [14,16,17]. In addition to their ability to combat drug-resistant bacterial strains, these compounds also possess properties that can inhibit fungi and fungal biofilms, including *C. auris* [17,18,19]. In this study, we investigated virulence factors and activity of ceragenins in seven *C. auris* strains isolated from five different hospitals in Türkiye. In addition, we assessed the effect of biofilm forms on drug efficacy. This is the first study investigating this issue in our country.

## 2. Materials and Methods

### 2.1. Isolates and Growth Conditions

The first isolated seven clinical *C. auris* strains from Türkiye, which were confirmed by DNA sequence analysis, were used in this study [7,20,21,22,23]. Isolates stored in sterile vials at −80 °C were cultured on Sabouraud Dextrose Agar (SDA-Difco, Sparks, MD, USA) plates and incubated at 35 °C overnight. *Candida parapsilosis* ATCC 22019, *Candida albicans* ATCC 90028, and *C. albicans* ATCC 10231 were used as quality control strains.

### 2.2. Virulence Factors

#### 2.2.1. Proteinase Activity

Production of proteinase was assessed according to a method described by Aoki et al. (1990) [24]. Special agar plates were prepared for the proteinase activity assay: 0.04 g MgSO_4_·7 H_2_O, 0.5 g K_2_HPO_4_, 1 g NaCl, 0.2 g dried extract of yeast, 4 g glucose, and 0.05 g bovine serum albumin mixed with 180 mL SDA. Inocula of 10 µL were streaked on plates and incubated at 37 °C for 6 days. *C. albicans* ATCC 10231 was used as positive control. Proteinase activity (Pz value) was calculated as the average ratio of the diameter of the colony to the total diameter of the colony plus the precipitation zone [25]. All experiments were carried out in triplicate.

#### 2.2.2. Phospholipase Activity

To determine phospholipase secretion of *C. auris* isolates, an SDA medium containing 14.6 g NaCl and 138 mg CaCl_2_ was prepared. After autoclave sterilization and cooling, 50 mL egg yolk was added, and inocula were spotted onto the medium. The inoculated plates were incubated at 37 °C for 4 days. *C. albicans* ATCC 10231 was used as a positive control [25]. The colony diameters and the precipitation zones were measured, and phospholipase activity was calculated for each isolate.

#### 2.2.3. Esterase Activity

The esterase activity of the isolates was assessed by the method described by Slifkin et al. (2000) [26]. Briefly, aliquots of a suspension from each isolate were carefully inoculated on a Tween-80 opacity test medium. The medium was prepared with the formulation of 10.0 g of bacteriological peptone, 5.0 g of NaCl, 0.1 g of CaCl_2_, 15 g of agar, and 1000 mL of distilled water. After autoclaving, the medium was cooled down to 50 °C, and 5 mL of Tween-80 was added into the medium. Incubation was performed at 37 °C for 10 days. The precipitation halo around the enzyme-expressing colonies was measured to determine esterase activity.* C. albicans* ATCC 10231 was used as positive control. Experiments were repeated three times.

#### 2.2.4. Hemolysin Production

The hemolytic activity was screened by the method described by Manns et al. (1994) [27]. Briefly, a standard inoculum containing approximately 10^8^ CFU/mL cells for each isolate was spotted onto sugar-enriched SDA medium supplemented with sheep blood and incubated at 37 °C for 48 h. *C. albicans* ATCC 90028, which is known for its strong hemolytic activity, was employed as the positive control. The presence of precipitation zone of hemolysis around the colony indicated hemolysin production. All experiments were carried out in triplicate.

#### 2.2.5. Evaluation of Virulence Factors

For all virulence factors, the diameter of colony and precipitation zone were measured, and the activity was calculated using the following formula:Pz: (colony diameter)/(colony diameter with precipitation zone diameter).

The results were classified as non-activity (Pz = 1), weak (0.64 < Pz < 0.99), and strong activity (Pz ≤ 0.63) [28].

#### 2.2.6. Biofilm Formation

A biofilm formation assay for seven *C. auris* strains was performed using a crystal violet assay [17]. *C. auris* inocula (1 × 10^7^ cfu/mL) in tryptic soy broth (TSB-BD, Heidelberg, Germany) containing 1% glucose (TSB-g) was added in six wells of 96-well flat-bottom tissue culture microplates for each isolate and incubated at 37 °C for 24 h. The next day, wells were washed with phosphate-buffered saline (PBS) three times, and methanol was added to the wells for 15 min for fixation. After removing methanol, a 1% crystal violet solution was added to dye biofilm mass for 5 min. Then, wells were washed with water to remove excess dye. The absorbed dye was dissolved with ethanol (96%). Absorptions (OD-optical density) were measured at 595 nm using a microtitre plate reader (EON-BioTEK Instruments, Winooski, VT, USA). Biofilm results were interpreted based on the following formula (c: negative control):OD ≤ ODc = No biofilm producer, ODc < OD < 2ODc = Low biofilm producer,
2ODc < OD < 4 ODc = Moderate biofilm producer, 4ODc ≤ OD = Strong biofilm producer.

### 2.3. Antifungal Susceptibility Tests

Broth microdilution assays were performed according to CLSI M27-A3 standard protocols [29]. Fluconazole, amphotericin-B, and caspofungin (Sigma-Aldrich, St. Louis, MO, USA) were used as antifungals. Ceragenins CSA-13, CSA-44, CSA-131, and CSA-138 were synthesized from a cholic acid scaffold technique as previously described [30]. Stock solutions of ceragenins from dry powders were prepared in water, and antifungals were prepared according to the manufacturers’ recommendations. Final concentrations of antimicrobials were prepared in Roswell Park Memorial Institute (RPMI, Sigma-Aldrich, St. Louis, MO, USA) medium supplemented with L-glutamine and buffered with morpholine propane sulfonic acid (MOPS; Sigma-Aldrich, St. Louis, MO, USA), before use. Briefly, the final concentration of yeast cell suspensions in RPMI medium was 1–5 × 10^3^ CFU/mL and distributed into a 96-well microtiter plate along with the compounds. The microtitre plates were incubated at 37 °C for 24–48 h. The lowest concentration inhibiting any discernible growth at 48 h was determined as the minimum inhibitory concentration (MIC) for amphotericin B. The lowest concentration associated with a 50% reduction in growth turbidity compared with the control well at 24 h was used as the MIC for caspofungin, and at 48 h was used as the MIC for fluconazole. Quality control (QC) isolate *C. parapsilosis* ATCC 22019 was included. During the experiments, the broth microdilution results of the quality control strains were determined at the range recommended by CLSI standard [29]. Experiments were performed in triplicate.

### 2.4. Antibiofilm Activity

The minimal biofilm eradication concentration (MBEC) assay was performed in 96-well polystyrene plates to evaluate the effect of antimicrobials on mature biofilm. Briefly, yeast cell suspensions at a 10^7^ cfu/mL density were inoculated into wells containing TSB-glucose (TSB-g) at 37 °C for 24 h. After the incubation, each well was washed thrice with 200 μL of sterilized PBS. Then, fresh medium containing various concentrations of antimicrobials (20–1280 μg/mL) was added to each well of the 96-well microtiter plate, followed by further incubation at 37 °C for 24 h. After exposure of antifungals to the biofilm, each well was gently washed three times with PBS. After adding PBS to all wells, the biofilm was detached by sonication in an ultrasonic cleaner (Branson Ultrasonic Cleaner; Branson Ultrasonics, Danbury, CT, USA) and vortexed. To determine CFU/mL, samples were taken from wells, diluted in PBS, and dropped onto SDA. After overnight incubation, colonies were counted. MBEC_99.9_ was defined as the lowest concentration to eradicate at least 99.9% of viable bacteria (a three-log reduction) in a biofilm compared to the growth controls.

## 3. Results

Upon testing the virulence characteristics of the strains employed in the study, it was observed that all strains exhibited comparable profiles, indicating a similarity in their pathogenic potential (Table 1). While proteinase and hemolysis activities were absent for all strains, phospholipase and esterase activities displayed weak levels of expression. Among the strains, two strains exhibited strong biofilm properties, while the other five strains exhibited moderate biofilm formation.

MIC results were interpreted according to the breakpoints recommended by the Centers for Disease Control and Prevention (CDC) [10]. Results are shown in Table 2. All strains were determined as resistant to fluconazole. The MIC values for amphotericin B against all strains were determined to be 1 µg/mL, whereas the MIC values for caspofungin ranged from 0.06 to 0.5 µg/mL. These findings indicate that the initial strains isolated from Türkiye resisted fluconazole but were susceptible to amphotericin B and caspofungin. MIC values of amphotericin B (0.25 µg/mL) and fluconazole (1 µg/mL) for quality control strain *C. parapsilosis* ATCC 22019 were within the expected MIC values.

The ceragenins displayed comparable antifungal activity to effective traditional antifungal agents. Among the ceragenins tested, CSA-131 and CSA-138 were identified as the most effective (Table 2). The MIC values for CSA-131 and CSA-138 were determined to be 1 µg/mL and 1–0.5 µg/mL, respectively. Furthermore, the MIC values for CSA-13 were found to be 8 µg/mL, while the MIC values for CSA-44 were determined as 4 µg/mL.

When evaluating the impact of antimicrobial agents on mature biofilms through biofilm studies, MBEC_99.9_ values for amphotericin B and caspofungin were observed to be in the range of 80–320 µg/mL and 40–320 µg/mL, respectively (Table 2). It was observed that the MBEC values of ceragenins were higher than the MBEC values of traditional antifungals. MBECs (µg/mL) for CSA-13, CSA-44, CSA-131, and CSA-138 were found to be 320–1280, 160–640, 80–1280, and 160–640, respectively.

## 4. Discussion

*C. auris* emerged as a human pathogen relatively recently but has become a significant health problem worldwide. However, our understanding of *C. auris* and associated infections is limited. A deeper understanding of this pathogen is important for effective management of associated infections. Consequently, there is a critical need to evaluate the pathogen’s virulence, drug resistance profile, and biofilm-forming capacity.

Proteinase, phospholipase, esterase, and hemolysin are hydrolytic enzymes known to be secreted from *Candida* spp. These enzymes play a crucial role in *Candida* spp. pathogenicity and aid in attaching to host tissues and damaging the host cell membrane. While phospholipases attack the phospholipids in the cell membrane as targets, esterase cleaves the ester bonds. Another virulence factor, the hemolysin enzyme, degrades the hemoglobin in the blood by extracting iron [31]. Through these enzymes, *Candida* spp. can invade mucous membranes and blood vessels while also evading the host’s immune response [32]. It has been determined that proteinases are associated with hyphae formation and increased adhesion in *Candida* spp., thus facilitating colonization in host tissues and disrupting defense proteins. Studies showed that proteinases play a critical role in infections caused by adhesion to mucosal membranes, such as oral and vaginal candidiasis [33]. The complete genome of *C. auris* has been unveiled in recent years; therefore, research into the genes and gene products responsible for its virulence and pathogenicity remains limited. Studies indicate that numerous genes exhibit either upregulation or downregulation during the synthesis of lytic enzymes, such as phospholipase [34]. In a case report by Kumar et al. (2015), proteinase, phospholipase, and hemolysin activity were determined in a *C. auris* strain isolated from vulvovaginal candidiasis [35]. In a study by Carvajal et al. (2021) examining the pathogenicity of 107 isolates associated with fungemia, proteinase activity was demonstrated in all strains [36]. Phospholipase activity was positive in 67.3% and hemolysin activity in 68.2% of the strains. Another study determined that 96% of the strains had adherence ability to epithelial cells and positive proteinase activity, while none showed phospholipase activity [37]. Larkin et al. (2017), in their study, found that the virulence factors of the tested strains were strain dependent, and 37.5% had phospholipase positivity, and 64% had proteinase positivity. They also reported that *C. auris* strains showed less adhesion to the catheter material than reference *C. albicans* strains [38]. *C. auris* can exhibit strain-dependent phenotypes, which may impact its pathogenicity [39,40]. Some strains, which form aggregates due to division without releasing daughter cells, display reduced pathogenicity compared to non-aggregated strains. Additionally, it has also been shown in the *Galleria mellonella* model that non-aggregate strains showed pathogenicity comparable to *C. albicans* [39]. In our study, similar phenotypic virulence characteristics were noted in all strains isolated from Türkiye. There was no activity detected for proteinase and hemolysin in all isolates. On the other hand, phospholipase and esterase activities were weak for all of them. Different virulence characteristics can be seen in *C. auris* strains tested in different studies. These variations can be explained by the fact that different isolates belong to different geographical clades. Also, *Candida* spp. may have different secretion patterns due to different isolation sources. For example, it has been shown that extracellular enzymes are proportional to the severity of the disease in *Candida* oral cavity infections occurring in diabetic patients and are highly expressed in *Candida* spp. isolated from these infections [31].

While considering CDC recommendations, three antifungal agents were included in the study to test antifungal susceptibility testing: fluconazole as triazole group antifungal, amphotericin B as polyene group antifungal, and caspofungin as echinocandin group antifungal. While the study results showed that the strains were resistant to fluconazole, they were found to be sensitive to amphotericin B and caspofungin. The CDC recommends using echinocandin as the initial treatment for *C. auris* infections. It has been stated that if there is no response to echinocandin treatment, liposomal amphotericin B can be considered [10]. Our study determined that seven strains isolated from Türkiye were susceptible to caspofungin and amphotericin B. Studies with *C. auris* susceptibility profiles have shown that *C. auris* is less sensitive to azoles than other antifungals. In a study, it was determined that *C. auris* fluconazole resistance was 90%. It has also been shown that resistance to amphotericin B is 8% and to echinocandins is 2% [41]. Similarly, another large-scale study found that almost all strains (99.8%) tested were resistant to fluconazole [42]. Azole resistance in *C. auris* can occur by different mechanisms. The most common is the mutation of the azole target gene, *ERG11*. In addition, efflux pump overexpression caused by a mutation in the transcription factor *TAC1* also results in azole resistance [43,44].

Biofilm formation is an important virulence factor for *C. auris*, as with other *Candida* spp. Biofilm formed by *C. auris* on artificial devices such as catheters and medical surfaces makes treatment more complex, causes drug resistance, and leads to hospital outbreaks [11]. The presence of multiple genes associated with biofilm formation has been confirmed. Notably, adhesin related genes *CSA1*, *IFF4*, *PGA26*, and *PGA52* have upregulated during the formation of *C. auris* biofilms. It has been observed that, as the biofilm matures, the genes encoding ABC transporter proteins like *CDR1, SNQ2*, and *YHD3* become activated. Additionally, agglutinin-like sequence (ALS) proteins ALS1 and ALS5 may play a role in contributing to biofilm formation [34,45]. Biofilm studies with *C. auris* from different geographical clades confirmed that the biofilm-forming feature of *C. auris* is a critical virulence factor [38,46]. A study of 11 strains isolated in a *C. auris* outbreak in Brazil showed that all strains formed a strong biofilm [46]. Larkin et al. (2017) determined that all of the *C. auris* isolates (n: 16) they tested formed a biofilm in their study. However, when they compared this biofilm layer with the *C. albicans* reference strain biofilm, they observed that *C. auris* formed a thinner biofilm layer than *C. albicans*. In addition, a more straightforward biofilm structure with less extracellular matrix composed of mainly yeast cells was noted [38]. Parallel to the other studies, our biofilm study with the strains isolated from Türkiye showed that all strains formed moderate to strong biofilms.

Biofilm formation is one of the most important causes of drug resistance. Biofilms are much more resistant to antifungal agents than planktonic cells. Chatzimoschou et al. (2020) determined the MIC values of various conventional antifungals against planktonic and biofilm forms of five strains, each belonging to a distinct clade of *C. auris*. The biofilm MIC was defined as the antifungal MIC corresponding to a 50% reduction in biofilm metabolism versus the control group’s mature biofilm. The results showed a very high increase in biofilm MICs of antifungals relative to *C. auris* planktonic cells [47]. In a previous study by Romero et al., the resistance of *C. auris* biofilms to antifungals was 512 times higher [48]. Similarly, according to the results of our study, the MBEC_99_ values of antifungals against biofilms were up to 1000 times higher than the MIC values against planktonic cells.

Reported widespread antifungal resistance for *C. auris* highlights the need to find new therapeutic alternatives. Ceragenins are one of these alternatives as membrane-active molecules, and their effectiveness against *C. albicans* or non-albicans strains has been demonstrated by multiple studies [17,18]. Microscopy studies have shown that ceragenins induce surface membrane damage in *Candida* cells [18]. Hashemi et al. (2018) investigated the activity of CSA-131 against 100 strains of *C. auris* from a CDC culture collection. The strains belonged to four different geographical clades and were isolated from different countries. They found that the MIC value of CSA-131 was 0.5–1 µg/mL [19]. Similarly, our study determined CSA-131 as one of the most effective ceragenins against Türkiye strains, while its MIC value was 1 µg/mL for all strains. In our study, CSA-138 was also noted as another ceragenin found to be as effective as CSA-131. Previous studies showed that CSA-138 has low MIC values against *C. albicans* and other non-albicans strains [16].

Similar to conventional antifungals, high concentrations of ceragenins are required to destroy the mature biofilm of *C. auris*. In a previous study, CSA-13 and CSA-131 have been shown to inhibit the growth of *C. albicans* biofilm. Additionally, its effect on DNA-mediated biofilm formation increased more in the presence of DNase [18]. High MIC values against biofilm cells reveal toxicity problems in treating biofilm infections. Therefore, using traditional antifungals and ceragenins at high doses (100–1000-fold) as antibiotic lock solutions can be considered to obtain antibiofilm activity [49]. Furthermore, it is advisable to test combinations of ceragenins with different agents, such as DNase, in order to reduce effective concentrations of ceragenins against *C. auris* biofilms. These aforementioned findings showed the importance of biofilm formation as a virulence factor because of the significant reduction in the sensitivity of antifungal agents. It can be argued that *C. auris* is a major threat in patients carrying devices such as catheters implanted in the body and on device surfaces in the hospital environment.

## 5. Conclusions

This study presents the first data on the phenotypic analysis of virulence factors and ceragenins efficacy in *C. auris* strains isolated from intensive care patients in Türkiye. To better understand *C. auris* strains’ virulence properties, it is recommended to conduct genotypic analysis and multicenter research with a larger series of strains. As an important finding, it was determined that leading ceragenins are as effective as traditional antifungals. Although these agents seem promising as an alternative treatment opinion, more studies are needed to clarify this issue.

## Figures and Tables

**Table 1 jof-09-01026-t001:** Virulence characteristic of *Candida auris* isolates.

	*Proteinase*		*Phospholipase*		*Esterase*		Hemolysin		Biofilm Formation	
	Pz Value	Activity	Pz Value	Activity	Pz Value	Activity	Pz Value	Activity	OD_595_	Activity
*C. auris 1*	0	No activity	0.830	Weak	0.761	Weak	0	No activity	0.451	Strong
*C. auris 2*	0	No activity	0.796	Weak	0.720	Weak	0	No activity	0.392	Strong
*C. auris 3*	0	No activity	0.776	Weak	0.700	Weak	0	No activity	0.260	Moderate
*C. auris 4*	0	No activity	0.787	Weak	0.692	Weak	0	No activity	0.208	Moderate
*C. auris 5*	0	No activity	0.804	Weak	0.755	Weak	0	No activity	0.228	Moderate
*C. auris 6*	0	No activity	0.792	Weak	0.720	Weak	0	No activity	0.261	Moderate
*C. auris 7*	0	No activity	0.833	Weak	0.723	Weak	0	No activity	0.276	Moderate
*C. albicans ATCC 10231*	0.350	Strong	0.412	Strong	0.500	Strong	ND	ND	ND	ND
*C. albicans ATCC 90028*	ND	ND	ND	ND	ND	ND	0.42	Strong	0.520	Strong
Media only									0.097	-

ND: not determined.

**Table 2 jof-09-01026-t002:** Antimicrobial activity results for *Candida auris* isolates.

	*Fluconazole*		*Amfoterisin*		*Kaspofungin*		CSA-13		CSA-44		CSA-131		CSA-138	
	MIC	MBEC	MIC	MBEC	MIC	MBEC	MIC	MBEC	MIC	MBEC	MIC	MBEC	MIC	MBEC
*C. auris 1*	>128	ND	1	160	0.06	160	8	1280	4	640	1	1280	1	640
*C. auris 2*	>128	ND	1	320	0.125	160	8	1280	4	160	1	640	1	640
*C. auris 3*	>128	ND	1	160	0.06	160	8	640	4	320	1	80	1	320
*C. auris 4*	>128	ND	1	80	0.06	160	8	640	4	320	1	80	1	160
*C. auris 5*	>128	ND	1	160	0.25	320	8	320	4	320	1	320	1	160
*C. auris 6*	>128	ND	1	80	0.06	40	8	160	4	640	1	640	0.5	320
*C. auris 7*	>128	ND	1	160	0.5	160	8	1280	4	640	1	160	0.5	640

ND: not determined.

## Data Availability

The data presented in this study are available on request from the corresponding author. The data are not publicly available due to ethical restrictions.

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
