# Peer review of "Phenotypic Investigation of Virulence Factors, Susceptibility to Ceragenins, and the Impact of Biofilm Formation on Drug Efficacy in Candida auris Isolates from Türkiye"

_jof, 2023, doi:10.3390/jof9101026_

Round 1

Reviewer 1 Report

I consider that the study is very interesting in using caragenins as an alternative for the in vitro treatment of Candida auris, but it is necessary:

1. Take into account that the title does not specify whether the virulence factors that were determined are phenotypic or genotypic, please review.

2. Enter the document number on which they based the susceptibility according to the CLSI guidelines.

3. When mentioning that the presence of phenotypic virulence factors was determined, it is important to establish, as part of the literature of the article, which genes could be involved so that these factors are evident, so I would ask the group of researchers who will investigate it and if they find it, they will add it to the article.

4. The study was done in one country, this greatly limits the results and the solidity of the study, but it is a good start to continue doing studies that include a larger series of these yeasts, possibly carrying out a multicenter study, the authors should consider this to place it among your conclusions or recommendations.

Reviewer 2 Report

In the Isolates and growth conditions section, add the temperature at which they were grown and also add how these yeasts were in stock (liquid nitrogen -180)?

In the Hemolysin Production section, specify why C. albicans strain ATCC90028 was used and not strain ATCC10231 as in the previous experiments.

On line 148 place the 7 (1x 107 cfu/mL) in superscript

On line 161 of the Antifungal susceptibility tests section, add which of the CLSI protocols was used.

On line 161 of the Antifungal susceptibility tests section, biofilm-embedded bacteria?

In table 1 instead of placing positive control and negative control, place the Candida strains used

In table 2 correct the names Amfoterisin and Kaspofungin

In lines 230 and 234 of the discussion remove the italics and apostrophe from spp.

In references write candida and candida species in italics
